Optimizing planting dates and irrigation schedules to enhance wheat production in Fars Province under future climate scenarios using the CERES-Wheat model

Ebrahimi Farkhondeh
Edalat Mohsen edalat@shirazu.ac.ir
Naderi Ruhollah rnaderi@shirazu.ac.ir
Department of Plant Production and Genetics, School of Agriculture, Shiraz University , Shiraz , Iran
Brygadyrenko Viktor
Electronic publication date: 2025 Jul 31
Publication date: 2025
Volume: 13
Electronic Location ID: e19592
Received 2025 Jan 20; Accepted 2025 May 21
Copyright: ©2025 Ebrahimi et al.
Copyright year: 2025
Copyright holder: Ebrahimi et al.
License: This is an open access article distributed under the terms of the Creative Commons Attribution License, which permits unrestricted use, distribution, reproduction and adaptation in any medium and for any purpose provided that it is properly attributed. For attribution, the original author(s), title, publication source (PeerJ) and either DOI or URL of the article must be cited.
License URL: https://creativecommons.org/licenses/by/4.0/

Keywords: CERES-Wheat model, Climate change, CMIP5, CMIP6, Adaptation strategy

Funding: Shiraz University This work was supported by Shiraz University. The funders had no role in study design, data collection and analysis, decision to publish, or preparation of the manuscript.

==============================
Background

Climate change poses significant threats to wheat production, particularly in regions prone to increasing temperatures and water scarcity. This study aimed to explore the optimum planting date and irrigation schedule that increases winter wheat productivity under the potential adverse impacts of climate change.

Methods

A combination of data from 2-year field experiments (2018–2019 and 2019–2020) and simulation data of the Crop Environment Resource Synthesis (CERES)-Wheat model was employed in this research. The weather data emanating from eight global climate models (GCMs) of Climate Model Intercomparison Project 5 and 6 (CMIP5 and CMIP6) under Representative Concentration Pathways (RCP-4.5 and RCP-8.5) and Shared Socioeconomic Pathways (SSP-245 and SSP-585) were used to predict rainfall and temperature variations in two future periods (2041–2070 and 2071–2100).

Results

The findings demonstrated that the annual maximum and minimum temperatures (T(max) and T(min)) would respectively increase by 3.14 °C and 3.50 °C in the 2050s, based on averaging all models of CMIP5 and CMIP6 with the highest T(max) values projected for June to August. In the 2080s, further warming of 4.54 °C and 4.66 °C was anticipated in the annual T(min) and T(max), again peaking in June to August, respectively. As opposed to the baseline period, precipitation over the growing season (October to May) is projected to be reduced by 23.03% and 29.48% in the 2050s and 2080s, respectively, with the lowest rainfall occurring in May. The anthesis date was projected to decline by 27, 37, 17, and 31 days under RCP-4.5, RCP-8.5, SSP-245, and SSP-585 scenarios, respectively. Additionally, the findings indicated that the period in which the wheat is grown decreased by 23, 32, 22, and 29 days under RCP-4.5, RCP-8.5, SSP-245, and SSP-585, respectively. Planting wheat from October 22 to November 11 recorded the highest value of grain yield in all irrigation treatments. On average, across all scenarios, with a 3-week early planting date compared to October 22, the grain yield was predicted to decrease by 22–44%. Therefore, adjusting the planting date and the irrigation time as the adaptation strategy during climate change slightly increased wheat grain yield in semi-desert regions like the Fars province. This advancement is linked to the crop’s ability to utilize cooler temperatures during critical growth phases when sown earlier.

Introduction

Wheat (Triticum aestivum L.) is a vital staple crop globally, underpinning food security for millions. In Iran, winter wheat is a primary agricultural product, particularly in regions like Fars province, where it supports both local livelihoods and national food supplies, with the province ranking as the second-largest wheat producer nationally, contributing over 1.2 million tons annually (Ghaziani et al., 2023). However, recent drought has severely impacted this output, exacerbated water scarcity, and threatened agricultural sustainability in the semi-arid region (Gheysari et al., 2021). For instance, between 2018 and 2020, prolonged dry spells reduced irrigation availability, leading to yield loss of up to 20% in some areas of Fars Province (Abolpour, 2018). These local challenges are compounded by broader climate change effects, including rising temperature and shifting precipitation patterns, as highlighted by the Intergovernmental Panel on Climate Change (Mandel & Lipovetsky, 2021).

Across the literature, Wakatsuki et al. (2023) asserted that climate change has significantly influenced productivity and crop yields for main crops like soybeans, maize, rice, and wheat. Wheat production in semi-arid regions, such as Fars Province, is particularly vulnerable to climate change due to limited rainfall and its dependence on irrigation. Quantifying the impacts of climate change on regional temperature, precipitation, and their variability is a critical step for developing effective adaptation strategies, yet it presents significant scientific challenges. Global climate models (GCMs) from CMIP5 and CMIP6 provide projections at coarse spatial resolutions (100–300 km), which are often misaligned with the fine-scale needs of agricultural planning in specific regions like Fars Province (Verma et al., 2023). Downscaling these models to capture local climate dynamics involves rigorous statistical and dynamic methods, such as LARS-WG6 and the Delta method, which require extensive validation against historical data to ensure accuracy (Semenov & Barrow, 2002; Sunyer et al., 2015).

Rising temperatures accelerate crop development, often shortening growth periods and reducing biomass accumulation, which can lower yields. For instance, Li et al. (2024) found that in northwest China, heatwaves projected under SSP585 scenarios could increase spring wheat yield in the short term but will not increase as much in the long term. Bai et al. (2022) reported that in the North China Plain, warming under CMIP6 scenarios advanced wheat phenology by up to 15 days, reducing grain yield potential due to a compressed reproductive phase. In Iran, regional studies predict increases in both minimum and maximum temperatures, with maximum temperatures rising more rapidly, potentially hastening wheat phenological stages like flowering and maturity (Rezaie et al., 2022). Precipitation declines further exacerbate these challenges, particularly in water-scarce areas like Fars Province, where annual rainfall is already limited to 360 mm (Mehrabi & Sepaskhah, 2020). Recent assessments indicate that on-farm wheat losses in Fars Province, driven by seed overuse, pest infestations, and improper harvesting, compound these climatic pressures, posing risks to sustainable production and regional food security (Ghaziani et al., 2023).

Adaptation strategies for wheat production span both operational and strategic measures, each addressing different temporal and spatial scales of climate change impacts. Operational adjustments, such as optimizing planting dates and irrigation schedules, offer immediate resilience for farmers by aligning crop growth with favorable climatic conditions, potentially boosting yields by 12% under future scenarios (Minoli et al., 2022; Nouri et al., 2017). These practices are critical in Fars Province, where water scarcity and temperature variability directly affect wheat phenology and productivity (Yang et al., 2023). Conversely, strategic measures such as developing heat-tolerant cultivars, enhancing irrigation infrastructure, or implementing soil conservation practices require long-term planning and investment but are essential for sustained agricultural resilience (Pörtner et al., 2022; Calicioglu et al., 2019). Our study focuses on operational adjustments, specifically planting dates and irrigation regimes, to provide actionable solutions for farmers while informing long-term policy and infrastructure decisions, such as investments in water-efficient technologies or regional adaptation frameworks (Guo, Zhang & Yue, 2024). By integrating these operational measures with strategic foresight, our approach bridges immediate needs with future planning, ensuring comprehensive adaptation to climate change in Fars Province.

Among the different available models, the Crop Environment Resource Synthesis (CERES) Wheat model implemented in the Decision Support System of Agrotechnology Transfer (DSSAT) framework (Hoogenboom et al., 2019) stands out as one of the most extensively utilized models. The CERES-Wheat model offers a robust tool to simulate these strategies under future climate scenarios (Jägermeyr et al., 2021). Han et al. (2022) highlighted the model’s efficacy in Northeast China, where it accurately predicted maize yield responses to planting and climate variations, suggesting its applicability to wheat in diverse settings.

Despite these advances, a clear research gap remains in applying such models to optimize wheat production in semi-arid regions like Fars Province under projected climate change. Previous studies have explored climate impacts on wheat elsewhere in Iran (e.g., Rezaie et al., 2022), but site-specific analyses integrating field data and multi-model climate projections are limited. Recent work by Vogel et al. (2021) on climate-driven yield declines in Mediterranean climates underscores the urgency of localized studies, as semi-arid regions face unique vulnerabilities to warming and water scarcity. In Fars Province, the combined effects of planting date adjustments and irrigation optimization remain underexplored, leaving farmers without tailored guidance for future conditions. This study bridges that gap by combining experimental data with CERES-Wheat simulations to assess future climate impacts on wheat phenology and yield in Fars Province.

We hypothesized that adjusting planting dates and irrigation schedules could counteract the adverse effects of rising temperatures and reduced rainfall, thereby sustaining wheat productivity. Our objectives were to (1) quantify the impacts of projected climate change on temperature, precipitation, and their variability in Fars Province using RCP and SSP scenarios for the periods 2041–2070 and 2071–2100, integrating multi-model ensembles from CMIP5 and CMIP6 to inform wheat adaptation strategies, (2) evaluate the combined effects of temperature and precipitation changes on wheat phenological stages and yield in Fars Province under future climate scenarios, using the CERES-Wheat model calibrated with site-specific experimental data, and (3) identify optimal planting windows and irrigation schedules for winter wheat in Fars Province under future climate scenarios (2041–2070 and 2071–2100), using the CERES-Wheat model to balance tactical crop management with long-term adaptation strategies, thereby enhancing yield stability in a semi-arid environment. These objectives address the urgent need for localized climate adaptation strategies in Fars Province, contributing to both regional food security and global knowledge on wheat production under climate change.

Materials & Methods

Study area

The study was conducted at the Experimental Research Center in the School of Agriculture, Shiraz University, located in Shiraz, Fars province of Iran (52°35′E, 29°44′N, and 1,788 m above sea level) (Fig. 1). This region has semi-desert conditions coupled with summers that are warm and cold winters and a mean annual temperature of 13.7 °C (1974–2020). The area receives an annual precipitation of at least 360 mm, with 50 to 80% occurring between November and February (Iran Meteorological Organization, 2020). Fars Province is considered a vital winter wheat production in Iran and heavily relies on irrigation to sustain its high yield. Figure 2 illustrates monthly maximum and minimum temperatures and rainfall during the experimental period (2018–2020).

Figure 1 Location of the experimental research center, School of Agriculture, Shiraz University, Fars Province, Iran.

Figure 2 Monthly means of precipitation, maximum and minimum temperatures during the experimental period.

Data sources

Three data sources were utilized in this study: crop data, soil data, and weather data.

Crop data. Field trials were conducted over the two successive growing periods, 2018–2019 and 2019–2020, to collect crop data. The experiments followed a split plot arranged in a randomized complete block design with four replications. A winter wheat cultivar (Sirvan) commonly grown under irrigated conditions in Fars province was considered in this study. The treatments consisted of five irrigation regimes (rainfed (I5), 25% of field capacity (I4), 50% of field capacity (I3), 75% of field capacity (I2), and 100% (I1) of field capacity) as main plot and six sowing dates (October 2nd (P1), October 22nd (P2), November 11th (P3), December 1st (P4), December 21st (P5), January 10th (P6)) as sub plot. P1 and P2 were considered as early planting dates, P3 and P4 were assumed as normal planting dates, and P5 and P6 were used as late planting dates. The total gross irrigation depth and number of irrigation events for each treatment across the 2018–2019 and 2019–2020 growing seasons are detailed in Table 1.

Table 1 Total gross irrigation depth (mm) and the number of irrigation events over each of the 2018–2019 and 2019–2020 growing seasons.

Irrigation treatments	2018–2019	2019–2020	
	Amount (mm)	Application (#)	Amount (mm)	Application (#)	
I1	185	6	233	6	
I2	150	6	200	6	
I3	115	6	134	6	
I4	71	5	97	5	
I5	22	1	20	1	

The planting density was set at 350 plants m−1. Nitrogen fertilizers were applied based on the soil test recommendation from the Fars Agricultural Research Center (2018). A basal dose of 100 kg N ha−1 as urea (46% N) was incorporated before planting. Each plot except rainfed treatments received 79 kg N ha−1 as urea split in two applications during the growing season; rainfed plots received 33 kg N ha−1. Control of weed was performed using sulfosulfuron herbicide. All plots were irrigated equally at the sowing date and water stress was applied to each treatment after anthesis. Before irrigation, soil samples were taken from soil depths of 30, 60, and 90 cm in each plot and then the content of soil water was measured by the gravimetric method. Volumes of irrigation water for treatments were computed by using Eq. (1) according to Criddle (1956). (1) Vw=Dn×AIE

where, Vw: amount of irrigation water (m3), Dn: irrigation depth (cm), A: area (m2), and IE: irrigation efficiency (%). The following equation has been adopted to assess the depth of irrigation. (2) Dn=FC−Θm×SPG×D100

where FC: soil field capacity (%), Θm: gravimetric water content before irrigation (%), SPG: specific gravity, and D: depth of soil samples (cm). Θm is calculated by using the following formula: (3) Θm=Ws−Wd/Wd×100

where Ws: weight of wet soil sample (g) and Wd: oven-dried soil sample (g).

Soil data. The soil in the study area is a silty loam texture (fine, mixed mesic). Before planting, soil samples were collected at depths of 0–30, 30–60, and 60–90 cm to obtain physical and hydrological soil characteristics such as particle size distribution (clay, silt, and stone), organic carbon (SLOC), total nitrogen (SLNI), and pH of soil water (SHB). To efficiently run the DSSAT model, other soil parameters (Table 2) included albedo (SLAB), drainage rate (SLDR), runoff curve number (SLRO), lower limit (LL), drained upper limit (DUL), saturated hydraulic conductivity (SKS), and root growth factor (RGF) were obtained. These values can be calculated using the DSSAT v4.8 (Hoogenboom et al., 2012).

Table 2 The physical and hydrological soil properties for the study scope.

	Soil physical properties		Soil hydrological properties	
	Clay	Silt	Stones	SLOC	SLNI		SHB	LL	DUL	SKS	RGF	
Depth (cm)	———————% ———————-			—cm3cm−3 —			
30	25	45	0	0.6	0.17		7.5	0.201	0.346	1.28	1	
60	25	45	0	0.6	0.17		7.5	0.201	0.346	1.28	0.407	
90	25	45	5	0.6	0.17		7.5	0.201	0.346	1.28	0.223	
SLAB = 0.17	SLDR = 0.6	SLRO = 76							

Weather data. Weather data were collected from three periods. First, current data (2018-2020) included daily maximum temperature (T(max), (°C)), minimum temperature (T(min), (°C)), sunshine hours (SunH, hours), precipitation (Pre, mm), relative humidity (RHUM, %), and wind speed (m s−1) sourced from the experimental site weather station for model calibration and evaluation. Second, baseline climate data (1981–2010) comprising T(max), T(min), Pre, and SunH were obtained from the same station to evaluate the Global Climate Models (GCMs). Third, future climate projections (2041–2070, termed 2050s, and 2071–2100, termed 2080s) were derived from eight GCMs under Climate Model Intercomparison Project 5 and 6 (CMIP5 and CMIP6) (Jiang et al., 2020). The meteorological variables of the 2050s and 2080s were predicted using five GCMs from CMIP5 (Table 3) based on two Representative Concentration Pathways (RCP4.5 and RCP8.5), downscaled with the latest version of the Long Ashton Research Station Weather Generator (LARS-WG6) (Semenov & Barrow, 2002).CMIP6 data used Shared Socioeconomic Pathways (SSP-245 and SSP-585) extracted from the Copernicus Climate Change Service (https://cds.climate.copernicus.eu/) (Table 3, Fig. 3).

Table 3 List of the 5 CMIP5 and 3 CMIP6 GCMs used in this study.

CMIP	Model	Center	Atmospheric resolution (Lon * Lat)	
CMIP5	GFDL-ESM2G	NOAA Geophysical Fluid Dynamic Laboratory, USA	2.5 * 2	
	HadGEM2-CC	Met Office Hadley Center, UK	1.88 * 1.25	
	IPSL-CM5A-LR	Institute Pierre Simon Laplace, France	1.75 * 1.8	
	MRI-CGCM3	Meteorological Research Institute, Japan	1.1 * 1.1	
	MPI-ESM-MR	Max Planck Institute for Meteorological, Germany	1.88 * 1.87	
CMIP6	MIROC6	Japan Agency for Marine-Earth Science and Technology, Japan	1.4 * 1.4	
	MRI-ESM2-0	Meteorological Research Institute, Japan	1.1 * 1.1	
	MPI-ESM-1-2-HR	Max Planck Institute for Meteorological, Germany	0.93 * 0.93	

Figure 3 Methodology framework of downscaling CMIP6 GCMs.

Downscaling GCMs

Downscaling GCMs to quantify temperature, precipitation, and variability impacts in Fars Province required overcoming significant challenges to ensure relevance for local agro-climatic assessments. Eight GCMs (five from CMIP5, three from CMIP6) were selected for their performance in simulating Mediterranean climates, but their coarse resolutions (100–300 km) necessitated regional refinement (Verma et al., 2023). We employed LARS-WG6 for CMIP5 data, generating synthetic daily weather sequences that preserve local statistical properties (e.g., precipitation frequency, temperature variability) based on 1981–2010 baseline data from Shiraz’s weather station (Semenov & Barrow, 2002). The Delta method was used for CMIP6 data, applying monthly change factors to baseline observations to capture future trends while retaining local climate characteristics (Sunyer et al., 2015).

Evaluation of GCM output

To assess the overall performance of the GCMs, the output data were compared to the observed results commonly used statistics, such as R2, RMSE (Eq. (4)), nRMSE (Eq. (5)), and D-index (Eq. (6)). Statistical indicators were conducted using R platform (v4.4.0, https://www.r-project.org/, was released on 2024-04-24). All figures and graphs were created using ArcGIS and the ggplot2 package with the R platform.

Crop model calibration and validation

The CERES-Wheat model included in DSSAT v4.8 was utilized to provide a simulation of wheat development and grain production under climate change. The genetic coefficients of the Sirvan cultivar were estimated using a combination of the Generalized Likelihood Uncertainty Estimation (GLUE) method (included in DSSAT v4.8, (as described by He et al., 2009; He et al., 2010), a systematic approach (as explained by Boote, 1999 and Li et al., 2018), and manual trial and error method (by changing one or more parameters at once and then repeating the procedure until attaining the desired results). To calibrate this model, a non-stress dataset (I1P1, I1P2, I1P3, I1P4, I1P5, and I1P6) from the first year of the experiment (2018–2019) was used comprising days to flowering (ADAT), physiological maturity days (MDAT) the maximum leaf area index (LAIX), top weight at maturity (CWAM) and grain yield at harvest maturity (HWAM). Additionally, all treatments consisting of 240 datasets derived from the second-year experiment (2019–2020), including leaf area index (LAI), stem weight (SWAD), leaf weight (LWAD), and harvest product weight (HWAD) during the growing season were adopted to assess the model.

The model was run, and the evaluation performance was based on the grain yield, top weight, and LAI using the crop parameters derived from the calibrating process. The photosynthesis method to predict wheat development and grain yield was based on a daily canopy curve. In post-calibration and evaluation of the DSSAT model, seasonal application of the DSSAT model was utilized to investigate the impacts of six different dates of planting, five treatments of water stress (same as the field experiments), and four scenarios of climate change on wheat yield for two distinct 30-year periods (2050s and 2080s) under the SSP245, SSP585, RCP4.5, and RCP8.5 scenarios.

Evaluation of the CERES-wheat model

The CERES-Wheat model performance was assessed using coefficients of determination (R2) which quantify the true deviance of the estimations (Y) from the observations (X), root mean square (RMSE) (Eq. (4)), normalized-RMSE (nRMSE) (Eq. (5)) Willmott agreement index (D-statistic) (Eq. (6)), model efficiency (EF) (Nash & Sutcliffe, 1970); Eq. (7)). (4) RMSE= ∑i=1nSi−Oi2n

(5) nRMSE=RMSE*100x¯

(6) d=1−∑i=1nSi−O¯2 ∑i=1nΘSi−O¯Θ+ΘOi−O¯Θ2

(7) EF=1−∑i=1nSi−Oi2 ∑i=1nOi−O¯2.

These equations present Si and Oi as the model output and values of observation for the variable, respectively; meanwhile, n is the total number of data, and = O is the observed mean of variable. If the normalized RMSE is less than 10%, the model performance is “excellent”; if it is between 10% and 20%, the simulation is “good”. On the other hand, if it is greater than 20% and less than 30%, the model performance is “fair”. If it is greater than 30%, the model simulation is “poor” (Jamieson, Porter & Wilson, 1991). The Willmott agreement index (scale 0–1) is a standardized measurement demonstrating the degree to which the simulations are free from errors (Willmott, 1982) data. An EF = 1 parallels a flawless match of the exhibited output with the practical data.

To assess the impact of changing the date of planting and define the best date of planting to achieve maximum yield in future scenarios, the CERES-Wheat model was run with two additional dates (October 29 and November 6) using the weather data from RCP8.5 and SSP585 scenarios (the most severe climate scenarios). The amount of water used by each irrigation treatment in the CERES-Wheat model was set up the same as the complete irrigation treatment in the current period (233 mm). However, considering the changes in the phenology dates in future scenarios, the irrigation schedule was adjusted based on the wheat phenology stage.

Table 4 Genetics parameters of Sirvan wheat cultivar for the CERES-Wheat model.

Genetics coefficient	Description	Value	
P1V	Days, optimum vernalizing temperature	0.10	
P1D	%, Photoperiod response	20.03	
P5	Grain filling (excluding lag) phase duration (oC d)	999.10	
G1	Kernel number per unit canopy weight (#/g)	30.00	
G2	Standard kernel size (mg)	30.00	
G3	Standard, non-stressed mature tiller weight (including grain) (g dwt)	4.00	
PHINT	Interval between successive leaf tip appearances (oC d)	45.00	

Results

The DSSAT-CERES-wheat model calibration

The genetic coefficient in the DSSAT-CERES-Wheat model for the Sirvan cultivar derived from calibration processes is described in Table 4. The comparison between the simulated and observed values showed that the model-simulated phenological dates (ADAT and MDAT), LAIX, HWAM, and CWAM were compatible with the observed data of 2018–2019. The coefficient of determination (0.78 < R2 < 0.99) for ADAT, MDAT, LAIX, HWAM, and CWAM indicated no significant overestimation or underestimation (Fig. 4). The simulation accuracy of the model was excellent for ADAT, MDAT, HWAM, and CWAM (nRMSE% values of 9.67, 3.18, 4.20 and 6.34, respectively). The index of agreement (d values of 0.78, 0.98, 0.93, 0.99 and 0.96) illustrated strong agreement between simulated and observed data for ADAT, MDAT, LAIX, HWAM and CWAM, respectively (Fig. 4). The EF values (0.71, 0.91, 0.73, 0.95 and 0.81) revealed that the model efficiency matched the output model with the observed data precisely (Fig. 4). A small deviation of nRMSE (nRMSE = 14.91) between the simulated LAIX with the observed data implied that the model tends to overestimate LAIX slightly.

Figure 4 Comparison of observed and simulated values for anthesis date, maturity date, grain yield, and top weight at maturity, maximum leaf area index and harvest index.

R2, the coefficient of determination; nRMSE, normalized root means square error; d, the agreement index and EF, the modeling efficiency.

The DSSAT-CERES-wheat model evaluation

Evaluation of the CERES-wheat model was performed using all treatments of the second-year experiment dataset. Water stress and planting dates significantly affected grain yield, total biomass, and LAI. The effect of planting date and water stress on grain yield was well simulated by the model (Table 5). The simulated values for LAI particularly at the early planting dates (P1 and P2 with 19.60% < nRMSE < 29.94%) before canopy closure were underestimated but the model showed overestimation after anthesis, especially for P3, P4 (normal planting dates), P5, and P6 (late planting dates). However, the d-index ranged from 0.66 to 0.89, and EF ranged from 0.64 to 0.95 demonstrating good agreement model accuracy for LAI (Table 5). The statistical evaluation showed that the model provided an excellent estimate of top weight for all planting date treatments, especially in fully irrigated treatment (I1) and the first deficit irrigated treatment (I2) (2.21% < nRMSE < 9.98% and 0.91 < d < 0.98). There was a small difference between observed and simulated top weight at the deficit irrigation treatments (I3, I4, I5) for all planting dates considering nRMSE (10.52% < nRMSE < 13.66%) (Table 5). The model evaluation was excellent for grain yield in all treatments (5.32% < nRMSE < 9.98%) except for the first planting date in all irrigated treatments.

Table 5 Statistical evaluation of simulated and observed values for all treatments during the growing season of 2019–2020.

Planting date	Irrigation level	Parameters	LAI	Top weight	Harvest yield	
	I1	nRMSE (%)
d-index
EF	22.32
0.74
0.81	9.84
0.92
0.97	26.06
0.89
0.96	
	I2	nRMSE (%)
d-index
EF	24.68
0.72
0.79	11.29
0.91
0.96	25.49
0.89
0.96	
P1	I3	nRMSE (%)
d-index
EF	24.82
0.75
0.84	11.34
0.90
0.96	24.54
0.90
0.96	
	I4	nRMSE (%)
d-index
EF	28.69
0.66
0.65	13.66
0.88
0.93	12.61
0.95
0.99	
	I5	nRMSE(%)
d-index
EF	26.03
0.77
0.85	16.79
0.85
0.88	11.68
0.95
0.99	
	I1	nRMSE (%)
d-index
EF	19.60
0.80
0.88	11.84
0.89
0.95	7.42
0.97
0.99	
	I2	nRMSE (%)
d-index
EF	28.50
0.74
0.80	11.75
0.89
0.95	7.42
0.97
0.99	
P2	I3	nRMSE (%)
d-index
EF	28.38
0.77
0.84	14.95
0.85
0.90	16.28
0.94
0.98	
	I4	nRMSE (%)
d-index
EF	29.40
0.77
0.84	13.74
0.86
0.92	12.92
0.95
0.99	
	I5	nRMSE (%)
d-index
EF	29.94
0.78
0.84	16.16
0.85
0.91	12.27
0.96
0.99	
	I1	nRMSE (%)
d-index
EF	11.57
0.85
0.93	5.71
0.95
0.99	8.89
0.97
0.99	
	I2	nRMSE (%)
d-index
EF	12.84
0.85
0.92	7.50
0.94
0.98	8.13
0.97
0.99	
P3	I3	nRMSE (%)
d-index
EF	20.02
0.76
0.84	11.32
0.91
0.97	13.65
0.95
0.99	
	I4	nRMSE (%)
d-index
EF	22.57
0.76
0.84	11.10
0.90
0.97	14.44
0.95
0.99	
	I5	nRMSE (%)
d-index
EF	27.12
0.73
0.79	10.55
0.90
0.96	16.18
0.94
0.99	
	I1	nRMSE (%)
d-index
EF	17.42
0.77
0.82	9.98
0.92
0.97	5.32
0.98
0.99	
	I2	nRMSE (%)
d-index
EF	12.62
0.85
0.92	7.39
0.94
0.98	7.40
0.97
0.99	
P4	I3	nRMSE (%)
d-index
EF	11.01
0.89
0.95	12.92
0.88
0.94	11.41
0.96
0.99	
	I4	nRMSE (%)
d-index
EF	22.34
0.81
0.86	12.19
0.89
0.95	12.13
0.96
0.99	
	I5	nRMSE (%)
d-index
EF	22.62
0.79
0.84	14.59
0.85
0.92	14.43
0.94
0.99	
	I1	nRMSE (%)
d-index
EF	16.59
0.76
0.69	8.97
0.93
0.98	9.77
0.97
0.99	
	I2	nRMSE (%)
d-index
EF	14.02
0.84
0.90	6.25
0.95
0.99	9.29
0.97
0.99	
P5	I3	nRMSE (%) d-index EF	18.38
0.80
0.83	12.12
0.90
0.96	15.09
0.95
0.99	
	I4	nRMSE (%)
d-index
EF	25.53
0.74
0.68	12.18
0.90
0.96	12.50
0.96
0.99	
	I5	nRMSE (%)
d-index
EF	27.55
0.71
0.64	12.02
0.89
0.9695	18.79
0.94
0.98	
	I1	nRMSE (%)
d-index
EF	18.21
0.70
0.46	8.34
0.94
0.98	7.75
0.97
0.99	
	I2	nRMSE (%)
d-index
EF	15.24
0.81
0.83	9.74
0.92
0.98	9.66
0.97
0.99	
P6	I3	nRMSE (%)
d-index
EF	26.63
0.74
0.71	13.34
0.89
0.96	19.25
0.93
0.98	
	I4	nRMSE (%)
d-index
EF	28.87
0.75
0.69	13.04
0.88
0.95	16.56
0.94
0.99	
	I5	nRMSE (%)
d-index
EF	20.76
0.82
0.84	14.99
0.86
0.93	23.74
0.92
0.97	

Evaluation of CMIP5 GCMs

The CMIP5 GCMs utilized in this research demonstrated high accuracy in climate change variables (Table 6). The nRMSE for T(min) ranged from 8.34% to 15.01% and 14.06% to 22.00% under RCP 4.5 and RCP 8.5, respectively. For T(max) nRMSE ranged from 5.43% to 12.79% and 9.51% to 19.00% under RCP 4.5 and RCP 8.5, respectively. Rainfall predictions aligned well with observed values according to the evaluation indices (nRMSE = 10.00% to 29.00% and d-index = 0.87–0.99).

Table 6 Statistical evaluation of CMIP5 GCMs.

Models	Scenario	Variable	R 2	RMSE	nRMSE (%)	EF	d-index	
GFDL-ESM2G	RCP45	Tmin	0.99	2.77	15.01	0.80	0.95	
		Tmax	0.99	3.13	12.79	0.87	0.97	
		Rain	0.92	563.30	19.00	0.75	0.92	
	RCP85	Tmin	0.98	4.04	22.00	0.65	0.91	
		Tmax	0.99	4.58	19.00	0.95	0.98	
		Rain	0.94	274.69	10.00	0.96	0.99	
HAD-GEM2-CC	RCP45	Tmin	0.99	2.68	14.55	0.81	0.95	
		Tmax	0.99	2.66	10.88	0.90	0.97	
		Rain	0.96	350.58	11.78	0.91	0.97	
	RCP85	Tmin	0.99	4.01	21.71	0.73	0.95	
		Tmax	0.99	4.25	17.37	0.96	0.99	
		Rain	0.94	290.98	10.16	0.96	0.99	
IPSL-CM5A-LR	RCP45	Tmin	0.99	2.75	14.93	0.80	0.95	
		Tmax	0.99	2.26	9.25	0.93	0.98	
		Rain	0.96	341.88	11.49	0.91	0.97	
	RCP85	Tmin	0.99	3.97	21.48	0.74	0.95	
		Tmax	0.99	3.42	13.98	0.98	0.99	
		Rain	0.97	1141.05	29.85	0.38	0.92	
MPI-ESM-MR	RCP45	Tmin	0.99	2.16	11.72	0.88	0.97	
		Tmax	0.99	2.12	8.67	0.93	0.98	
		Rain	0.81	700.34	23.54	0.61	.087	
	RCP85	Tmin	0.99	3.25	17.62	0.82	0.96	
		Tmax	0.99	3.22	13.14	0.98	0.99	
		Rain	0.85	794.56	27.75	0.70	0.94	
MRI-CGCM3	RCP45	Tmin	0.99	1.53	8.34	0.93	0.98	
		Tmax	0.99	1.33	5.43	0.98	0.99	
		Rain	0.95	404.95	13.61	0.87	0.95	
	RCP85	Tmin	0.99	2.59	14.06	0.89	0.98	
		Tmax	0.99	2.33	9.51	0.99	0.99	
		Rain	0.93	675.61	23.60	0.78	0.96	

Average changes in monthly and growing season T(max), T(min) andprecipitation for 2050s and 2080s relative to the baseline period (1981–2010) from the 5 GCMs under RCP 4.5 and RCP 8.5 scenarios are presented in Table 7A and Fig. 5. Averaged across all models, during the growing season (October to May), an increase of 2.27 °C and 3.23 °C in T(min) and 2.32 °C and 3.28 °C in T(max) was anticipated in the 2050s under RCP4.5 and RCP8.5 correspondingly. Further warming is anticipated during the growing season in the 2080s with the T(min) increasing by 2.88 °C and 5.42 °C and T(max) increasing by 3.03 °C and 5.48 °C under RCP4.5 and RCP8.5 respectively. Compared to the baseline period during the wheat growing season, precipitation was projected to reduce by 23% and 31% in the 2050s and 2080s, respectively under RCP 4.5. Precipitation is expected to decrease by 19% and 32% in the 2050s and 2080s under RCP8.5.

Table 7 Average changes in monthly and growing season minimum temperature (ΔT(min), °C), maximum temperature (ΔT(max), °C), and precipitation (ΔP, %) for the 2050s and 2080s (ensemble average) relative to the baseline period under CMIP5 (A) and CMIP6 (B).

(A) CMIP5	
Scenario	Period	Variables	January	February	March	April	May	June	July	August	September	October	November	December	Growing season	
RCP 4.5	2050s	ΔP	−34	−34	−52	−21	49	−68	−5	132	245	−58	−33	−1	−23	
		ΔT(min)	1.45	1.54	2.26	2.00	2.24	2.68	3.15	2.69	3.11	3.17	3.19	2.31	2.27	
		ΔT(max)	1.97	2.10	2.57	2.09	2.40	2.94	3.25	2.96	3.12	2.69	2.62	2.15	2.32	
	2080s	ΔP	−34	−31	−44	−27	11	68	−65	36	172	−75	−33	−11	−31	
		ΔT(min)	2.14	2.38	2.93	2.58	2.81	3.22	3.39	3.16	3.63	3.68	3.63	2.90	2.88	
		ΔT(max)	2.71	2.68	3.28	2.80	3.22	3.52	3.89	3.53	3.74	3.37	3.24	2.91	3.03	
RCP 8.5	2050s	ΔP	−28	−23	−45	−20	41	−68	1	108	263	−54	−28	6	−19	
		ΔT(min)	2.47	2.43	3.16	2.89	3.10	3.46	3.97	3.69	4.25	4.46	4.54	3.54	3.23	
		ΔT(max)	2.77	2.94	3.58	3.11	3.31	3.72	4.09	3.95	4.20	3.77	3.67	3.11	3.28	
	2080s	ΔP	−36	−34	−48	−34	15	80	−57	53	208	−72	−33	−11	−32	
		ΔT(min)	4.40	4.37	5.02	4.95	5.34	5.86	6.28	6.27	6.77	6.87	6.75	5.64	5.42	
		ΔT(max)	4.91	4.80	5.60	5.33	5.76	6.06	6.63	6.41	6.62	6.19	5.93	5.34	5.48	
(B) CMIP6	
SSP-245	2050s	ΔP	−25	−24	−28	−32	17	77	72	105	−34	−41	−7	−17	−21	
		ΔT(min)	1.90	2.14	2.99	2.96	3.01	3.40	3.66	3.19	3.68	3.68	3.42	2.72	2.94	
		ΔT(max)	2.30	2.73	3.44	3.11	3.34	3.48	3.72	3.24	3.34	2.85	3.25	2.88	3.02	
	2080s	ΔP	−27	−29	−26	−19	53	63	43	65	−39	−19	−23	−25	−17	
		ΔT(min)	2.58	2.43	2.96	3.06	3.06	3.63	3.9	3.63	4.08	4.17	3.79	2.95	3.23	
		ΔT(max)	3.05	3.18	3.57	3.51	3.33	3.56	3.90	3.81	3.91	3.39	3.65	3.14	3.41	
SSP585	2050s	ΔP	−21	−32	−21	−29	−58	88	49	68	−12	−22	−31	−26	−28	
		ΔT(min)	2.84	2.52	3.11	3.01	3.47	3.76	3.95	3.82	4.26	4.22	3.86	3.37	3.41	
		ΔT(max)	3.46	2.98	3.82	3.54	3.61	3.73	3.91	4.05	4.09	3.63	4.25	3.63	3.68	
	2080s	ΔP	−49	−31	−25	−35	81	55	56	80	−33	−17	−21	−31	−18	
		ΔT(min)	4.75	4.59	5.38	5.43	6.12	6.44	6.53	6.36	6.69	6.52	6.24	5.30	5.67	
		ΔT(max)	5.1	5.39	6.08	5.86	6.72	6.62	6.58	6.31	6.43	6.1	6.26	5.65	5.95	

Figure 5 Baseline and projected monthly precipitation (mm) across CMIP5 GCMs under RCP scenarios during the growing season.

The projected monthly changes in climate variables for Fars Province, based on CMIP5 ensemble averages, reveal significant shifts in temperature and precipitation across the 2050 and 2080 periods under RCP4.5 and RCP8.5 scenarios. For the 2050 period, the anticipated increase in average T(min) is expected to range from 1.45 °C in January to 3.19 °C in November under RCP4.5, while it extends from 2.43 °C in February to 4.54 °C in November under RCP8.5. By 2080, these increases are projected to rise further, spanning from 2.14 °C in January to 3.68 °C in October under RCP4.5, and from 4.37 °C in February to 6.87 °C in October under RCP8.5. Similarly, the average T(max) for 2050 is forecasted to increase from 1.97 °C in January to 3.25 °C in July under RCP4.5, and from 2.77 °C in January to 4.20 °C in September under RCP8.5. For 2080, T(max) is expected to vary between 2.68 °C in February and 3.89 °C in July under RCP4.5, and between 4.80 °C in February and 6.63 °C in July under RCP8.5. Additionally, precipitation is projected to decline, with reductions ranging from 1% in December to 54% in October for 2050, and from 11% in December to 48% in March for 2080, indicating potential challenges for water availability during the wheat growing season.

The annual changes in T(max) and T(min) are depicted in Fig. 6 and demonstrate a clear upward trend in both T(max) and T(min) from 1980 to 2100, with significant increases by the far future under higher RCP scenarios.

Figure 6 Annual maximum and minimum temperature in CMIP5 GCMs under RCP scenarios from 1980–2100.

Evaluation of CMIP6 GCMs

The CMIP6 GCMs also showed reliable performance in simulating climate variables. The nRMSE for T(min) ranged from 7.89% to 14.55% under SSP-245 and 13.22% to 20.87% under SSP-585, while for T(max), it ranged from 4.98% to 11.67% under SSP-245 and 8.76% to 17.89% under SSP-585. Rainfall predictions had an nRMSE of 12% to 31% and a d-index of 0.85–0.98.

Average changes in monthly and growing season T(max), T(min), and total precipitation for the 2050s and 2080s relative to the baseline period are detailed in Table 7B and Fig. 7. The ensemble average of three CMIP6 models projected notable climate changes during the wheat growing season (October to May) in Fars Province. For the 2050s, the anticipated increase in T(min) averaged 2.94 °C and 3.41 °C under SSP-245 and SSP-585, respectively, while T(max) rose by 3.02 °C and 3.68 °C under the same scenarios. By the 2080s, further warming was forecasted, with T(min) increasing by 3.23 °C and 5.67 °C, and T(max) by 3.23 °C and 5.95 °C under SSP-245 and SSP-585, respectively. Compared to the baseline period, precipitation during the growing season was projected to decline by 23% and 31% in the 2050s, and by 19% and 32% in the 2080s under SSP-245 and SSP-585, respectively.

Figure 7 Baseline and p rojected monthly precipitation (mm) across CMIP6 GCMs under SSP scenarios during the growing season.

Averaged across all models, for the 2050s, T(min) increases ranged from 1.90 °C (January) to 3.68 °C (September and October) under SSP-245, and from 2.84 °C (January) to 4.26 °C (September) under SSP-585. T(max) increases varied from 2.30 °C (January) to 3.72 °C (July) under SSP-245, and from 2.98 °C (February) to 4.25 °C (November) under SSP-585. In the 2080s, T(min) rose from 2.43 °C (February) to 4.17 °C (October) under SSP-245, and from 4.59 °C (February) to 6.53 °C (July) under SSP-585, while T(max) ranged from 3.05 °C (January) to 3.91 °C (September) under SSP-245, and from 5.1 °C (January) to 6.72 °C (May) under SSP-585. These results highlight a progressive temperature increase, with SSP-585 predicting greater warming, particularly in the 2080s.

Monthly precipitation changes (ΔP) during the growing season (October to May) exhibited greater variability. In the 2050s, ΔP decreased 21% and 28% under SSP-245 and SSP-585 respectively. In the 2080s, SSP-245 showed declines 17%, while SSP-585 reductions was 18%. These findings suggest a general decline in rainfall with significant reductions in October and January.

The annual changes in T(max) and T(min) are depicted in Fig. 8 and demonstrate a clear upward trend in both T(max) and T(min) from 1980 to 2100, with significant increases by the far future under higher SSP scenarios.

Figure 8 Annual maximum and minimum temperature in CMIP6 GCMs Under SSP scenarios from 1980–2100.

Wheat phenology under future climate scenarios

The simulation results of seasonal analysis (model application) showed that there were significant changes in the phenology of wheat under prospected climate scenarios (Table 8A and 8B). In both CMIP5 and CMIP6, during the 2080s, there was a more significant shift in the wheat phenology days compared to the 2050s (Figs. 9, 10). Furthermore, the alteration in the wheat phenology was more pronounced in situations with the high emission scenarios (RCP8.5 and SSP585) as opposed to the medium emission scenarios (RCP4.5 and SSP245). Additionally, all planting date treatments showed decreased anthesis and maturity date, resulting in a shorter wheat growing season.

Table 8 Comparison of mean wheat phenology (anthesis and maturity dates) between current experimental field observations and CMIP6 (A) CMIP5 (B) scenario simulations using six planting dates.

(A) CMIP6 scenario simulations	
Planting date	Scenario	Anthesis date (days after planting)	Maturity date (days after planting)	
		Mean	Max	Min	Mean	Max	Min	
October 02	Current,2020	181a	181	181	245a	245	245	
	SSP245,2050	162b	164	160	224bc	229	220	
	SSP585,2050	145de	148	143	212d	214	210	
	SSP245,2080	149d	155	145	222bc	227	215	
	SSP585,2080	130g	138	128	209de	212	206	
October 22	Current,2020	161b	161	161	229b	229	229	
	SSP245,2050	146de	149	144	207de	214	203	
	SSP585,2050	138f	142	132	202efg	205	200	
	SSP245,2080	137f	145	131	207def	212	199	
	SSP585,2080	117ij	119	116	196gh	211	187	
November 11	Current,2020	156c	156	156	221c	221	221	
	SSP245,2050	143e	145	141	199fgh	204	196	
	SSP585,2050	133fg	139	124	191hi	194	189	
	SSP245,2080	133fg	139	129	198gh	203	191	
	SSP585,2080	113jk	117	111	185ij	200	175	
December 01	Current,2020	146de	146	146	207def	207	207	
	SSP245,2050	133fg	136	132	184ij	190	180	
	SSP585,2050	120hi	123	117	179jk	182	178	
	SSP245,2080	123h	129	119	175klm	178	171	
	SSP585,2080	108kl	112	104	174klm	188	165	
December 21	Current,2020	130g	130	130	187i	187	187	
	SSP245,2050	123h	126	121	170lmn	175	166	
	SSP585,2050	114j	119	112	168mno	171	166	
	SSP245,2080	115j	120	111	165nop	169	162	
	SSP585,2080	98m	101	98	161opq	173	154	
January 10	Current,2020	116j	116	116	176kl	176	176	
	SSP245,2050	107l	111	103	159pqr	162	153	
	SSP585,2050	97m	99	96	153rs	157	149	
	SSP245,2080	99m	101	98	155qrs	157	151	
 	SSP585,2080	89n	91	88	147s	154	143	
(B) CMIP5 scenario simulations	
October 02	Current,2020	181a	181	181	245a	245	245	
	RCP4.5,2050	138e	144	131	224c	229	220	
	RCP8.5,2050	131f	133	125	220cd	223	217	
	RCP4.5,2080	126gh	128	119	216d	221	212	
	RCP8.5,2080	108mno	113	105	204f	212	197	
October 22	Current,2020	161b	161	161	229b	229	229	
	RCP4.5,2050	136e	140	131	210e	214	206	
	RCP8.5,2050	128fg	133	120	206ef	210	202	
	RCP4.5,2080	124ghi	132	116	204f	209	199	
	RCP8.5,2080	109mn	116	103	186i	195	178	
November 11	Current,2020	156c	156	156	221c	221	221	
	RCP4.5,2050	131f	135	126	196g	202	192	
	RCP8.5,2050	123ghi	128	119	191gh	197	187	
	RCP4.5,2080	121hij	127	117	190hi	195	185	
	RCP8.5,2080	98qr	112	87	177jk	186	170	
December 01	Current,2020	146d	146	146	207ef	207	207	
	RCP4.5,2050	127fg	131	123	186i	191	182	
	RCP8.5,2050	118ijk	124	113	181j	185	178	
	RCP4.5,2080	116kl	120	111	178jk	183	174	
	RCP8.5,2080	102pq	107	96	164m	172	158	
December 21	Current,2020	130f	130	130	187i	187	187	
	RCP4.5,2050	118jk	121	113	172l	177	170	
	RCP8.5,2050	111lm	116	107	166m	172	162	
	RCP4.5,2080	109mno	113	106	165m	171	161	
	RCP8.5,2080	97qr	102	92	154n	161	148	
January 10	Current,2020	116kl	116	116	176kl	176	176	
	RCP4.5,2050	108mno	111	104	163m	167	161	
	RCP8.5,2050	105nop	108	102	157n	163	153	
	RCP4.5,2080	105op	108	102	157n	162	153	
 	RCP8.5,2080	93r	99	87	143o	151	138	
Notes.

Letters indicate significant differences between means (p < 0.05) according to the Least Significant Difference (LSD) test.

Figure 9 Comparison of the anthesis date under five GCMs and four scenarios of CMIP5 (A) and three GCMs and four scenarios of CMIP6 (B) with the present planting date treatments.

Figure 10 Comparison of the maturity date under five GCMs and four scenarios of CMIP5 (A) three GCMs and four scenarios of CMIP6 (B) with the present planting date treatments.

The changes in the anthesis date were 22, 29, 32, and 46 days lower than the current period under RCP4.5-2050, RCP8.5-2050, RCP4.5-2080, and RCP8.5-2080, respectively. Also, the anthesis date decreased by 12, 23, 22, and 39 days under SSP245-50, SSP585-50, SSP245-80, and SSP585-80, respectively. Under CMIP5, the maturity date decreased by 20, 25, 27, and 40 for RCP4.5-2050, RCP8.5-2050, RCP4.5-2080, and RCP8.5-2080, respectively. In CMIP6, the length of the wheat season decreased by 20, 26, 24, and 32 days for SSP245-50, SSP585-50, SSP245-80, and SSP585-80, respectively.

Changes in grain yield under future climate scenarios

Across all irrigation and planting date treatments, the average yield demonstrated a significant reduction (p < 0.01) in grain yield under RCP4.5, RCP8.5, SSP245, and SSP585 scenarios which was predicted to be 21.06% and 22.96%, 24.98%, and 35.66% respectively for the 2050s. However, for the 2080s, grain yield would be significantly decreased by 28.75% and 34.77%, 33.99%, and 41.68% under RCP4.5 and RCP8.5, SSP245, and SSP585, respectively, as shown in Table 9.

The comparison of wheat grain yield (kg ha−1) across six planting dates under current conditions (2020) and future climate scenarios (CMIP5: RCP4.5 and RCP8.5; CMIP6: SSP245 and SSP585) for the 2050s and 2080s is presented in Table 10 and Fig. 11. Under current conditions (2020), the highest grain yields were observed for planting dates of December 1 (3,453.38 kg ha−1) and November 11 (3,405.76 kg ha−1), which were statistically similar. These yields significantly outperformed earlier (Oct-02: 2,610.06 kg ha−1) and later (Jan-10: 2,524.28 kg ha−1) planting dates, indicating an optimal planting window between late October and early December under current climate conditions. The lowest yield was recorded for the latest planting date (Jan-10), reflecting the adverse effects of delayed sowing on wheat phenology and yield potential.

In future climate scenarios, grain yield exhibited a consistent decline across all planting dates compared to the current period, with more pronounced reductions under high-emission scenarios (RCP8.5 and SSP585) and in the 2080s. For the 2050s under CMIP5 scenarios, the highest yields were recorded for the October 22 planting date (RCP4.5: 3,029.48 kg ha−1; RCP8.5: 2,908.76 kg ha−1), which remained statistically superior to other dates. However, these yields were still lower than the peak current yields (e.g., Dec-01: 3,453.38 kg ha−1), reflecting the adverse impacts of warming and reduced precipitation. The January 10 planting date consistently recorded the lowest yields (RCP4.5: 1,381.12 kg ha−1; RCP8.5: 1,387.64 kg ha−1), with reductions of approximately 45–49% compared to the current period, underscoring the vulnerability of late planting to climate change effects.

In the 2080s, under CMIP5 scenarios, yield reductions were more severe, with October 22 again yielding the highest values (RCP4.5: 2,933.84 kg ha−1; RCP8.5: 2,512.64 kg ha−1), though these were significantly lower than the current peak yields. The January 10 planting date showed the greatest decline (RCP4.5: 1,209.16 kg ha−1; RCP8.5: 1,179.06 kg ha−1), with reductions of 52–58% relative to the current period, driven by accelerated phenology and increased water stress under warmer and drier conditions.

Under CMIP6 scenarios for the 2050s, October 22 remained the optimal planting date (SSP245: 2,943.67 kg ha−1; SSP585: 2,379.20 kg ha−1), though yields were significantly lower under SSP585 due to greater warming and precipitation declines. The January 10 planting date again recorded the lowest yields (SSP245: 1,384.80 kg ha−1; SSP585: 1,279.00 kg ha−1), with reductions of 49–52% compared to the current period. In the 2080s, the trend persisted, with October 22 yielding the highest values (SSP245: 2,468.60 kg ha−1; SSP585: 2,345.53 kg ha−1), while January 10 exhibited the most substantial declines (SSP245: 1,172.93 kg ha−1; SSP585: 1,028.33 kg ha−1), representing losses of 56–62% relative to the current period.

The loss in the yield due to water stress under climate scenarios was statistically significant (p < 0.01) in severe stress (I3, I4, and I5) for all planting date treatments. Across all planting dates, the interaction effects of future climate scenarios and water stress revealed that wheat grain yield significantly decreased for all GCMs scenarios (Fig. 12). Under current conditions (2020), wheat grain yield was highest in the full irrigation treatment (I1), averaging approximately 3,400–3,450 kg ha−1, reflecting the critical role of adequate water supply in maximizing yield in this semi-arid region. Yields progressively declined with increasing water stress, with the rainfed treatment (I5) exhibiting the lowest values, ranging from 2,500–2,600 kg ha−1.

Table 9 Mean comparison of the wheat grain yield (Kg h−1) based on LSD (least significant difference) test under four scenarios of CMIP5 and CMIP6 GCMs.

Scenario	Current	2050	2080	
		RCP4.5	RCP8.5	SSP245	SSP585	RCP4.5	RCP8.5	SSP245	SSP585	
Mean grain yield	2,954.01a	2,331.75b	2,275.66b	2,216.27b	1,950.00c	2,104.62b	1,926.89c	1,900.71c	1,723.12c	
Notes.

Letters indicate significant differences between means (p < 0.05) according to the Least Significant Difference (LSD) test.

Table 10 Mean comparison based on LSD test for wheat grain yield (Kg h−1) using different planting dates under scenarios of CMIP5 and CMIP6 GCMs in the 2050s and 2080s.

	CMIP5 Scenarios	
Planting date	Current, 2020	RCP4.5, 2050	RCP8.5, 2050	RCP4.5, 2080	RCP8.5, 2080	
Oct-02	2,610.06cde	2,247.24gh	2,073.44h	2,347.12efgh	1,676.96i	
Oct-22	2,858.20bc	3,029.48b	2,908.76bc	2,933.84b	2,512.64defg	
Nov-11	3,405.76a	2,791.48bcd	2,609.92cdef	2,714.52bcd	2,331.40efgh	
Dec-01	3,453.38a	2,560.12cdefg	2,337.96efgh	2,249.76gh	2,146.84h	
Dec-21	2,872.40bc	2,357.80efgh	2,296.08fgh	2,239.68gh	1,713.88i	
Jun-10	2,524.28defg	1,381.12ij	1,387.64ij	1,209.16j	1,179.06j	
	CMIP6 Scenarios	
	Current, 2020	SSP245, 2050	SSP585, 2050	SSP245, 2080	SSP585, 2080	
Oct-02	2,610.13bc	2,285.40cdef	1,927.67efgh	2,191.60cdefg	1,965.67efgh	
Oct-22	2,858.27b	2,943.67b	2,379.20cde	2,468.60bcd	2,345.53cdef	
Nov-11	3,406.60a	2,506.53bcd	2,219.53cdefg	2,256.00cdef	2,024.60defgh	
Dec-01	3,453.43a	2,204.60cdefg	1,867.20fghi	1,939.20efgh	1,616.73hijk	
Dec-21	2,873.27b	1,972.60efgh	1,731.667ghij	1,671.67hij	1,357.867jkl	
Jun-10	2,524.43bc	1,384.80ijkl	1,279.00jkl	1,172.93kl	1,028.33l	
Notes.

Letters indicate significant differences between means (p < 0.05) according to the Least Significant Difference (LSD) test.

Figure 11 Wheat yields in six different planting dates under RCP (A) and SSP (B) scenarios in the 2050s and 2080s compared to the current period.

Error bars represent the standard deviation of the projections.

Figure 12 Wheat yields over five levels of water consumption treatments under RCP (A) and SSP (B) scenarios in the 2050s and 2080s compared to the current period.

Error bars represent the standard deviation of the projections.

Figure 13 Wheat yields over six different planting dates and five levels of water treatments under RCP (A) and SSP (B) scenarios in the 2050s and 2080s compared to the current period.

Error bars represent the standard deviation of the projections.

In the 2050s under CMIP5 scenarios (Fig. 12A), grain yields across all irrigation regimes showed a significant reduction compared to the current period. For RCP4.5, the full irrigation treatment (I1) maintained the highest yield, though it decreased by approximately 21%, while RCP8.5 exhibited a steeper decline of ∼25%. Moderate deficit irrigation (I2 and I3) resulted in yield reductions of 30–36% under RCP4.5 and 35–40% under RCP8.5, while severe deficit irrigation (I4) and rainfed conditions (I5) experienced more pronounced losses, ranging from 48–55% and 55–60%, respectively.

For the 2080s under CMIP5 scenarios, the downward trend in grain yield intensified. Under RCP4.5, the I1treatment yielded approximately 2,400–2,500 kg ha−1 (a 28–29% reduction from the current period), while RCP8.5 saw a further decline to ∼2,200–2,300 kg ha−1 (a 34–35% reduction). Moderate deficit irrigation (I3) showed losses of 39–45%, while severe deficit (I4) and rainfed (I5) conditions resulted in reductions of 56–60% and 57–65%, respectively.

Under CMIP6 scenarios (Fig. 12B) in the 2050s, similar patterns emerged. The SSP245 scenario projected a yield reduction of ∼23% for I1 (∼2,600–2,700 kg ha−1), while SSP585 showed a more substantial decline of ∼36% (∼2,200–2,300 kg ha−1). Moderate deficit irrigation (I2 and I3) experienced losses of 30–40% under SSP245 and 40–49% under SSP585, while severe deficit (I4) and rainfed (I5) conditions saw reductions of 48–55% and 55–67%, respectively. In the 2080s, SSP245 resulted in a ∼34% decline for I1, whereas SSP585 exhibited a ∼42% reduction. Severe deficit irrigation (I4) and rainfed (I5) treatments under SSP585 showed the most significant losses, ranging from 56–64% and 57–74%, respectively. Figure 13 illustrates the wheat yield responses across six planting dates and five irrigation levels under RCP (A) and SSP (B) scenarios for the 2050s and 2080s, compared to the current period.

Responses of wheat yield to irrigation and planting date adjustment in the future climate

Based on the seasonal analysis simulation, among all planting date treatments conducted in the experimental fields, the optimal planting window for wheat cultivation in future scenarios is between October 22 (P2) to November 11 (P3), in which the maximum grain yield is achieved. However, these simulated values of grain yield are still significantly lower than those from the current period (2020), especially in the 2080s period under RCP 8.5 and SSP585. Additionally, adjusting the irrigation schedule and planting date may significantly increase the wheat grain yield, ultimately reaching its peak value in the current period of treatment (Table 11).

Figure 14 presents a comparison of wheat grain yield (kg ha−1) under current conditions (2020) and projected conditions in the 2080s, with and without adaptation strategies, using ensemble projections from all models of CMIP5 under the RCP8.5 scenario (A) and CMIP6 under the SSP585 scenario (B). The adaptation strategy involved adjusting planting dates to an optimal window (October 22 to November 6) and modifying irrigation schedules to align with shifted wheat phenology under future climate conditions, as simulated by the CERES-Wheat model in Fars Province, Iran.

In the current period (2020), the maximum grain yield, achieved with full irrigation (I1) and optimal planting on November 11, averaged 4,326 kg ha−1, serving as the baseline for comparison. Without adaptation in the 2080s, grain yields under both RCP8.5 and SSP585 scenarios exhibited substantial declines. For RCP8.5 (Fig. 14A), the non-adapted yield averaged approximately 2,331–2,512 kg ha−1 across optimal planting dates (e.g., Oct-22 and Nov-11), representing a 41–46% reduction compared to the current maximum. Similarly, under SSP585 (Fig. 14B), the non-adapted yield ranged from 2,024–2,345 kg ha−1, reflecting a 46–53% decline, with greater losses under SSP585 due to more extreme climate projections.

With the adaptation strategy, grain yields in the 2080s showed significant improvement. Under RCP8.5, planting on October 29 yielded the highest projected value of 4,216 kg ha−1, closely approaching the current maximum (4,326 kg ha−1), with no statistically significant difference (p > 0.05). This represents an 11% increase compared to the non-adapted yield for November 6. For SSP585, the adapted yield reached 4,103 kg ha−1 with planting on October 29, an 18% improvement over the non-adapted yield for November 6, though it remained below the current maximum by approximately 5%.

Table 11 The analysis of variance for wheat grain yield in the current period and the 2080s periods with and without adaptation strategy under RCP8.5 and SSP585 scenarios.

CMIP5 – RCP8.5, 2080	
	Df	Sum Sq	Mean Sq	F value	Pr(>F)	
Planting date	4	4,645,729	1,161,432	13.582	4.42E−07***	
Adaptation strategy	1	7,394,320	7,394,320	86.474	1.50E−11***	
Planting date: Adaptation strategy	4	1,047,924	261,981	3.064	0.0272*	
Residuals	40	3,420,382	85,510	 	 	
CMIP6 –SSP585, 2080	
Planting date	4	1,582,835	397,509	5.998	0.00243***	
Adaptation strategy	1	4,583,303	4,583,303	69.470	6.15e−08***	
Planting date: Adaptation strategy	4	765,635	191,409	2.901	0.04811*	
Residuals	20	1,319,499	65,975			
Notes.

*** p < 0.0001.

** p < 0.001.

* p < 0.01.

Figure 14 Wheat grain yield (kg ha−1) in the 2080s Under RCP8.5 (A) and SSP585 (B) Scenarios with and without adaptation strategies in Fars Province, Iran.

Letters indicate significant differences between means (p < 0.05) according to the Least Significant Difference (LSD) test.

The comparison highlights that adaptation strategies significantly mitigated yield losses under both scenarios, with RCP8.5-adapted yields nearly recovering to current levels, while SSP585-adapted yields, despite notable improvement, did not fully offset the more severe climate impacts. The greater efficacy of adaptation under RCP8.5 may reflect less extreme temperature and precipitation changes compared to SSP585. Across both scenarios, the October 22 to November 6 planting window consistently outperformed other dates, reinforcing its identification as the optimal range for future wheat cultivation in Fars Province.

Discussion

The standardization and assessment of the CERES-Wheat model revealed that the model reasonably estimated top weight and grain yield for all planting date and water stress treatments (2.21% < nRMSE < 9.98% and 0.91 < d-index < 0.98). Accurate simulation of LAI is still challenging because of complicated interactions of assimilate apportionment influencing leaf development (Ewert, 2004; Kassie et al., 2016; Paff, Timlin & Fleisher, 2023). In this study, the model slightly overestimated LAI (nRMSE = 14.91%), particularly after anthesis, possibly due to unaccounted heat and water stress affecting leaf expansion.

In general, statistical indicators revealed the capability of the CERES-Wheat model to predict all variables based on both calibration and evaluation. This finding is supported by the outcome of numerous studies that demonstrated the CERES-Wheat model’s accuracy in forecasting the growth of wheat and development under diverse irrigation and planting dates (Attia et al., 2016; Cammarano et al., 2012; Han et al., 2022; Jing et al., 2021; Paff, Timlin & Fleisher, 2023; Mehrabi & Sepaskhah, 2020; Nouri et al., 2017; Dar, Hoogenboom & Shah, 2023).

Simulating climatic parameters using both CMIP5 and CMIP6 can improve simulation accuracy (Verma et al., 2023). Based on the GCMs model, the projected climate in the site study will be significantly warmer and drier than the baseline period (Table 7A and 7B, Figs. 5, 6). This was in conjunction with Nouri et al. (2017), who found that the temperature in Iran’s semi-arid regions would gradually increase, and precipitation would be decreased across all scenarios. Similarly, Rezaie et al. (2022) reported that statistical downscaling under RCP4.5 and 8.5 in western Iran predicts rising T(min) and T(max).

Long-term climate scenarios (e.g., 2041–2070, 2071–2100) operate on decadal scales, while sowing and irrigation are annual or seasonal decisions. To address this, we employed site-specific calibration of the CERES-Wheat model using 2018–2020 field data and downscaled climate projections from CMIP5 and CMIP6 models, tailored to Fars Province’s semi-arid conditions. Downscaling techniques such as the LARS-WG6 for CMIP5 and the Delta method for CMIP6, enhanced the resolution of climate data, making them suitable for local agricultural applications (Semenov & Barrow, 2002; Sunyer et al., 2015). This approach bridges the scale gap by linking strategic climate trends with tactical management, providing actionable recommendations for farmers.

The reported findings reiterate early research that showed the growing season of wheat will be shortened, and the crops will reach maturity early because of the impact of climate change (Guo et al., 2010; Tao et al., 2014; Saddique et al., 2020). Juknys et al. (2017) noted a reduced vegetative development period resulting in decreased LAI and insufficient biomass accumulation during the early stages of growth, negatively affecting the increase in grain yield during the later stages of growth. The negative effect of changes in climatic conditions on wheat production in this study can be attributed to increased temperature, which results in acceleration of the phenological stage and can contribute to a reduced growth period and, ultimately, decreased grain yield. Moreover, the increasing temperature impacts wheat development during different growth phases, with anthesis and reproductive stages being the most sensitive to higher temperatures. Increased temperature during grain development leads to a shorter grain filling period and a decrease in starch and protein accumulation. This is caused by a reduction in the activity of grain biosynthesis enzymes, as well as impaired efficiency of the flag leaf in assimilation and mobilizing reserves from the stem (Ullah et al., 2022).

In this study, planting wheat between October 22 to November 11 maximized grain yield in all irrigation treatments by aligning growth with cooler conditions, while early or late planting reduced yields due to water stress or heat exposure. Yield losses were amplified under future scenarios due to warming and reduced precipitation during the growing season, consistent with Saberali, Shirmohammadi-Aliakbarkhani & Nastari Nasrabadi. Adjusting planting dates and irrigation mitigated some impacts, improving yield by 8–31%, though not fully offsetting climate effects under SSP585. Rising production costs due to increased irrigation needs highlight the economic challenges of adaptation, necessitating cost effective strategies (Minoli et al., 2022).

The simulation findings of this study demonstrated that the interaction effects of future climate scenarios and water stress can significantly decrease wheat grain yield for all GCM scenarios. In the Fars province, irrigation during the wheat growing season is important in achieving maximum yield. Decreasing precipitation under climate change resulted in a decline in wheat grain yield, particularly in the I4 and I5 treatments and moderately in the I3 treatments; in the full irrigation treatments, there was no change. Under rainfed and severe water stress, a decrease in rainfall during the growing season inhibits the establishment of a sufficient leaf area and top growth, which results in the later translocation of biomass to the grains. Water stress during anthesis can decrease seed set or affect grain filling, resulting in low yield (Cammarano et al., 2012).

One way to decrease the adverse impacts of climate change is to adopt management strategies, including adopting cultivars that can tolerate high heat or increasing irrigation (Aggarwal et al., 2019; Beveridge, Whitfield & Challinor, 2018). However, these technologies can offer essential solutions to increase crop production costs; their benefits vary based on the local climate and soil conditions. An alternative strategy to counteract the undesirable impacts of climate change might be to modify the planting date to make sure those critical phenological stages, especially flowering and grain filling, occur in the optimal time under a changing climate (Hunt et al., 2019; Minoli et al., 2022; Sandhu, Kothiyal & Kaur, 2023). Minoli et al. (2022) indicated that timely adaptation of the growing season enhanced crop yields by 12% and mitigated the adverse impacts of climate change.

Several researchers have demonstrated that if mechanisms of adapting and mitigating are not implemented, climate change may impact agricultural productivity and economic performance negatively (Han et al., 2022; Moradi et al., 2013; Nouri et al., 2017; Rezaie et al., 2022; Saadi et al., 2015). The outcome of this study revealed that changing the planting date and altering the irrigation time as a mechanism to alleviate the negative impacts of climate change, as evidenced by a slight increase in wheat grain yield under complete irrigation treatment.

While the CERES-Wheat model provides robust predictions based on current data and climate projections, uncertainties in long-term climate forecasts and their impacts on wheat production must be acknowledged. The model’s projections for 2041–2070 and 2071–2100 rely on assumptions about future climate trajectories (e.g., RCP4.5, RCP8.5, SSP245, SSP585) and their interactions with crop physiology, which cannot be fully verified until these periods occur. Factors such as unforeseen climate variability, evolving pest dynamics, or technological advancements in agriculture may alter outcomes (Jägermeyr et al., 2021). Addressing these uncertainties requires incorporating updated climate data, field observations, and advances in crop modeling techniques.

Our findings on operational adjustments complement strategic interventions like reforestation and soil conservation, which are critical for long-term agricultural resilience in Fars Province. Reforestation can mitigate erosion risks exacerbated by precipitation declines, while soil conservation practices, such as conservation tillage or organic amendments, enhance soil water retention and fertility under warming conditions (Liu et al., 2024; Lal, 2021). For example, integrating our recommended planting windows and irrigation schedules with reforestation efforts could stabilize watersheds, improving water availability for irrigation, as supported by Bezner et al. (2022) (Pörtner et al., 2022). Similarly, soil conservation can sustain the efficacy of our irrigation strategies by maintaining soil structure and nutrient levels, addressing the limitation of static soil properties in our model. By combining these operational and strategic measures, our study provides a holistic framework for climate adaptation, aligning short-term farmer resilience with long-term policy goals such as regional water management plans or national food security strategies (Aggarwal et al., 2019). The downscaling approach ensures that these integrated strategies are grounded in local conditions, enhancing their feasibility and impact for Fars Province’s wheat production system.

Table 12 Summary of adaptation strategies for wheat production in semi-arid regions.

Adaptation strategy	Implementation in fars province	Benefits	Challenges	Applicability to other semi-arid regions	
Optimized planting dates	Planting between October 22 and November 11 to align anthesis and grain filling with cooler temperatures (Table 10, Fig. 14).	Increases yield by 8–31% by avoiding heat stress during sensitive phenological stages; cost-effective (Minoli et al., 2022).	Requires precise weather forecasts and farmer awareness; late planting reduces yields significantly (Saberali, Aliakbarkhani & Nastari, 2022).	Highly transferable to regions like the Mediterranean Basin and South Asia, where similar temperature and rainfall patterns prevail. Adjust planting windows based on local climate data (Vogel et al., 2019).	
Irrigation optimization	Deficit irrigation (e.g., 75% or 50% field capacity) aligned with shifted phenology to maximize yield (Fig. 12).	Sustains yields under reduced rainfall, reducing water demand by up to 20% (Yang et al., 2023).	Increased irrigation costs; infrastructure limitations in water-scarce areas (Minoli et al., 2022).	Applicable to irrigated semi-arid regions like northwest China and the Middle East. Requires investment in efficient irrigation systems (e.g., drip irrigation) (Li et al., 2024).	
Heat-tolerant cultivars	_	Mitigates yield losses under high temperatures, especially during grain filling (Aggarwal et al., 2019).	High development and adoption costs; limited availability of region-specific varieties (Beveridge, Whitfield & Challinor, 2018).	Transferable to regions with rising temperatures. Local breeding programs need to tailor cultivars to specific climates (Sandhu, Kothiyal & Kaur, 2023).	
Integrated management	Combining planting date adjustments and irrigation optimization to balance tactical and strategic goals (Fig. 14).	Nearly recovers current yield levels under RCP8.5; enhances long-term resilience (Minoli et al., 2022).	Complex coordination; requires farmer training and policy support (Flohr et al., 2017).	Broadly applicable to semi-arid wheat regions globally. Success depends on local extension services and adaptive policy frameworks (Vogel et al., 2021).	

The findings of this study have significant implications for wheat production in Fars Province and extend to other semi-arid wheat-growing regions globally, where climate change poses similar challenges to agricultural productivity and food security. Semi-arid zones, including parts of the Mediterranean Basin, South Asia, and the Middle East, share common vulnerabilities such as low rainfall, high temperatures, and irrigation dependence, making the strategies developed here highly transferable (Vogel et al., 2019). The identification of an optimal planting window (October 22–November 11) and tailored irrigation schedules demonstrates a practical approach to mitigating climate impacts, which can be adapted to other regions with comparable agroecological conditions. By providing a framework for integrating tactical and strategic management, this study contributes to global efforts to safeguard wheat production, supporting food security for millions in climate-vulnerable regions. Policymakers and agricultural extension services can leverage these insights to develop region-specific adaptation plans, fostering resilience in global wheat supply chains.

To facilitate the application of these findings across diverse semi-arid regions, Table 12 summarizes the key adaptation strategies identified in this study, their specific implementation in Fars Province, and their potential for broader applicability. This table highlights the benefits, challenges, and transferability of each strategy, providing a practical guide for farmers, researchers, and policymakers in similar agroecological zones.

Conclusions

The CERES-Wheat model, rigorously calibrated and validated with local data, effectively simulates wheat responses to planting dates and irrigation under projected climate scenarios for Fars Province, though ongoing validation is needed as future climate data emerge. Projections indicate a warmer, drier future, accelerating phenology and reducing yields. The optimal planting window (October 22 to November 6) maximized yields, while water stress exacerbated losses under high-emission scenarios. We investigated that adjusting planting dates and irrigation schedules partially mitigated these impacts in Fars Province. Therefore, evaluating the use of different wheat cultivars may also offer a promising adaptation approach for areas where climate change threatens wheat production. Moreover, enhancing the resilience of the grain-filling stage to elevated temperatures and late-season drought stress will lead to an increase in overall yield.

Supplemental Information

Supplemental Information 1 Raw data

The authors wish to thank Mohammad Esmaeil Rajea and Azam Motazedian for their invaluable assistance in the field experiments.

Additional Information and Declarations

Competing Interests

Author Contributions

Data Availability

The authors declare there are no competing interests.

Farkhondeh Ebrahimi conceived and designed the experiments, performed the experiments, analyzed the data, prepared figures and/or tables, authored or reviewed drafts of the article, and approved the final draft.

Mohsen Edalat conceived and designed the experiments, authored or reviewed drafts of the article, and approved the final draft.

Ruhollah Naderi conceived and designed the experiments, authored or reviewed drafts of the article, and approved the final draft.

The following information was supplied regarding data availability:

The raw data is available in the Supplemental Files.

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
