# Peer review of "Optimizing planting dates and irrigation schedules to enhance wheat production in Fars Province under future climate scenarios using the CERES-Wheat model"

_PeerJ, doi:10.7717/peerj.19592_

## Round 0.1 · original submission · Major Revisions

Dear authors, I ask you to carefully correct the manuscript in accordance with the reviewers' fundamental comments. I ask you to show the data in the figures, where possible, in the form of a box analysis (median, first and third quartiles, minimum and maximum values).

·

Basic reporting

Language: The English language should be improved to ensure that the audience clearly understands your findings. Some of your sentences are too long, lack smooth transitions, and contain grammatical errors. Additionally, certain parts require rephrasing for better clarity. For example, lines 57-58 could be rewritten as: "Climate change affects the growth and production of wheat, with its impact varying depending on the location of the study."
Title: I have an alternative title suggestion as ”Optimizing Planting Dates and Irrigation Schedules to Enhance Wheat Production in Fars Province Under Future Climate Scenarios Using the CERES-Wheat Model”
Introduction and background: Your introduction lacks smooth transitions between sentences, affecting coherence and logical flow. Additionally, it does not clearly highlight the research gap. References are needed in lines 66 and 69. Sentences in lines 72–74 require rephrasing for clarity and conciseness, and the text in lines 82–85 appears to have a different font, which should be standardized.
Moreover, the introduction should be restructured to provide a clear rationale, starting with an overview of wheat crop production status, the challenges faced, and the solutions proposed by scholars. It should then address the specific challenges of wheat production in your region or research site. Finally, clearly state your research hypothesis and objectives.
Structure: Your manuscript follows the structure of the journal.
Figures: The figures are relevant and of good quality. However, the unit labeling lacks proper subscripting. Ensure that all units are correctly formatted with appropriate subscripts for clarity and accuracy.
Raw data: Thank you for providing the raw data. I think your data is descriptive.

Experimental design

Research questions: In my view, the research questions need to be rewritten and more clearly articulated. Although the gaps are listed in the introduction, they lack logical flow and are not well expressed. Improving coherence and clarity will enhance the overall structure and impact of the study.
Materials and Methods: Please provide references for the data listed in lines 102–107 and for the fertilizer recommendations used. Additionally, ensure that all units are correctly formatted, such as:
Temperature (°C) in line 107
Proper superscripting of units (e.g., m⁻¹ in line 121 and ha⁻¹ in lines 122–124)
It is better to use "soil texture" instead of "soil type" in line 138, as soil texture specifically refers to the relative proportions of sand, silt, and clay particles, while soil type is a broader term.
Lines 177–182 form a single, lengthy sentence. Please divide it into shorter, clearer sentences for better readability.
Overall, the methodology lacks details on procedures such as how leaf area index, leaf weight, and harvested product weight were measured. Providing a clear description of these measurement techniques will enhance the clarity and reproducibility of the study

Validity of the findings

Impact and novelty: I recommend reviewing the findings of Saberali et al. (2022) ( https://doi.org/10.1007/s00704-022-04005-8) conducted in Iran and comparing findings with your results in the discussion section. This comparison will help highlight the novelty and significance of your findings more effectively.
Results: You did well. However, please check the correctness of units in lines 266–269 to ensure proper formatting. In line 321, replace "However" with a more positively connecting word such as "Moreover" or a similar term, as the previous sentences do not contrast with the idea presented in lines 321–323.
Discussion: In lines 380–381, you stated that accurately simulating LAI is challenging based on previous studies. It would be better to also explain how this aspect was in your study. Additionally, please review the logical flow of your discussion to ensure clarity and coherence.
Conclusions: The conclusion is generally well written, but minor revisions (e.g., lines 491–493) are needed for clarity and conciseness.
4. General comments: The title should be modified for clarity and precision. Additionally, ensure a smooth flow of ideas throughout the manuscript by improving transitions between sections. Rephrase long and unclear sentences to enhance readability and comprehension.

Additional comments

No comment

·

Basic reporting

In the abstract, you mentioned the highest temperatures you will reach during the forecast periods without indicating which month they were in.
Which of the months had the lowest rainfall? Was it within the plant growth period?
In Table 5, no units were mentioned.

Experimental design

no comment

Validity of the findings

no comment

Reviewer 3 ·

Basic reporting

The text of the article and grammar should be generally improved. There are currently statements that are ambiguous and far from clear. Certain parts of the manuscript need to be re-structured. There are figures and tables that need to be added to the presentation of results.

Experimental design

The experimental design is satisfactory and generally well-explained in the manuscript.

Validity of the findings

Some supporting information regarding the main findings is currently missing, but can be provided and the presentation of results can and should be improved. See the detailed comments in my general report.

Annotated reviews are not available for download in order to protect the identity of reviewers who chose to remain anonymous.

---

## Round 0.2 · Major Revisions

Dear Dr. Naderi, I ask you to correct the manuscript in accordance with the reviewer's fundamental comments. I hope that the new version of the article will be approved for publication.

·

Basic reporting

The authors have considered my previous comments; therefore, I have no additional comments

Experimental design

The authors have considered my previous comments; therefore, I have no additional comments

Validity of the findings

The authors have considered my previous comments; therefore, I have no additional comments

Additional comments

No comment

·

Basic reporting

Global climate change is a significant challenge for all of humanity. Climate change poses significant risks to agricultural production. Increasing temperatures and decreasing precipitation are very threatening for the seven-decade area, as investigated in this article. Forecasting production conditions over a range of several decades is important for identifying measures that have a significant lag time to achieve their results. For example, assessing the need for reforestation as a factor in preventing the growing risks of erosion induced by global climate change. It should be noted that optimal planting dates and irrigation regimes are elements of operational (rather than strategic) crop management, so the need for a strategic forecast to solve such problems is questionable due to the mismatch between the spatial and temporal scales of global climate change forecasts and the rhythm of operational crop management.

The authors consider the following research objectives:
(1) to assess regional climate change using RCP and SSP scenarios up to 2100 - this is a mechanical procedure of projecting existing scenarios that already cover the area. Such a goal cannot be considered as a scientific research goal - it is a trivial component of research methods.
(2) assess the impacts of temperature and precipitation changes on wheat growth stages and yield is also a mechanical procedure involving existing models implemented in routine software packages. The use of the authors' experimental data in this procedure makes this objective a routine scientific and technical procedure.
(3) to determine the optimal planting dates and irrigation regimes to enhance wheat production under future climates - this objective indicates a fundamental mismatch between the spatial and temporal scales of the phenomena being linked and the authors' attempt to compare elements of tactical (definitive) and strategic crop management, which is a mistake.

Experimental design

EXPERIMENTAL DESIGN
The experimental design is presented flawlessly. In principle, this is expected, since it is a routine arithmetic procedure. The authors should highlight the design features that provide an opportunity to obtain new scientific value and will be of interest to other researchers and practitioners for further application. Attention should also be paid to possible limitations and shortcomings of the procedure, which may limit the accuracy of the results.
Three data sources were used in this study: crop data, soil data, and weather data. However, a significant drawback of the approach is that the authors assume by default that soil properties will not change in response to global climate change. This is not the case, and the contribution of soil to changing crop growing conditions needs to be considered. By the way, soil properties are quite inertial, so measures should be planned now to correct soil conditions in a few decades.

Validity of the findings

Conclusions: ‘The CERES-Wheat model is effective in predicting wheat responses to planting dates and irrigation under future climates in Fars Province’ is not true and is only an assumption of the authors. It will be possible to verify this conclusion only after climate change occurs.
‘The optimal planting window (October 22 to November 6) maximised yields’ - how does this differ significantly from the current conditions for growing this crop? Will there be a shift in the optimal window, will it expand or contract?
In general, the value of a forecast that the planting time for a particular crop will need to be shifted in a particular province in a few decades is highly questionable. The scientific interest, especially for a wide range of readers, is also questionable.

On the positive side, I should note the almost impeccable quality of the manuscript, which is why I do not propose to reject it.
I suggest that the authors consider the problem more broadly and find aspects that will make the results more scientific and practical, which will attract the interest of other scholars.

---

## Round 0.3 · accepted · Accept

Dear Dr. Naderi, I am pleased to inform you that this article has been accepted for publication. I hope that you will continue to submit such high-quality articles to our journal for publication in the future.

·

Basic reporting

My comments were adressed

Experimental design

My comments were adressed

Validity of the findings

My comments were adressed

Additional comments

I have no adational comments

·

Basic reporting

The authors have implemented all the recommendations of the reviewer. The quality of the manuscript has been significantly improved.

Experimental design

The authors have implemented all the recommendations of the reviewer. The quality of the manuscript has been significantly improved.

Validity of the findings

The authors have implemented all the recommendations of the reviewer. The quality of the manuscript has been significantly improved.

Additional comments

The authors have implemented all the recommendations of the reviewer. The quality of the manuscript has been significantly improved. I recommend the article for publication